# Imaging of the Internal Structure of Permafrost in the Tibetan Plateau Using Ground Penetrating Radar

**Yao Wang [1,2], Zhihong Fu [1,2,\*], Xinglin Lu [1,2], Shanqiang Qin [1,2] 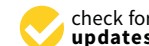, Haowen Wang [1,2] and Xiujuan Wang [1,2]**

1 State Key Laboratory of Power Transmission Equipment and System Security and New Technology, Chongqing University, Chongqing 400044, China; 20161101003@cqu.edu.cn (Y.W.); 20191101379@cqu.edu.cn (X.L.); shqiangqin@cqu.edu.cn (S.Q.); 20097482@cqu.edu.cn (H.W.); 20111101002@cqu.edu.cn (X.W.)

2 School of Electrical Engineering, Chongqing University, Chongqing 400044, China

\* Correspondence: fuzhihong@cqu.edu.cn; Tel.: +86-130-6235-2738

**Abstract:** The distribution of the permafrost in the Tibetan Plateau has dramatically changed due to climate change, expressed as increasing permafrost degradation, thawing depth deepening and disappearance of island permafrost. These changes have serious impacts on the local ecological environment and the stability of engineering infrastructures. Ground penetrating radar (GPR) is used to detect permafrost active layer depth, the upper limit of permafrost and the thawing of permafrost with the season's changes. Due to the influence of complex structure in the permafrost layer, it is difficult to effectively characterize the accurate structure within the permafrost on the radar profile. In order to get the high resolution GPR profile in the Tibetan Plateau, the reverse time migration (RTM) imaging method was applied to GPR real data. In this paper, RTM algorithm is proven to be correct through the groove's model of forward modeling data. In the Beiluhe region, the imaging result of GPR RTM profiles show that the RTM of GPR makes use of diffracted energy to properly position the reflections caused by the gravels, pebbles, cobbles and small discontinuities. It can accurately determine the depth of the active layer bottom interface in the migration section. In order to prove the accuracy of interpretation results of real data RTM section, we set up the three dielectric constant models based on the real data RTM profiles and geological information, and obtained the model data RTM profiles, which can prove the accuracy of interpretation results of three-line RTM profiles. The results of three-line RTM bears great significance for the study of complex structure and freezing and thawing process of permafrost at the Beiluhe region on the Tibetan Plateau.

**Keywords:** ground penetrating radar; reverse time migration; Tibetan Plateau; permafrost active layer; internal structure

## 1. Introduction

The Tibetan Plateau is known as the "third pole of the world." Its average altitude is higher than 4500 m, which gives it the highest and most complex terrain of a plateau with permafrost regions in the world. The character of Tibetan Plateau permafrost has obvious vertical zoning. With the increase of altitude, the freezing depth of permafrost obviously ascends. Compared with North America and Russia's arctic permafrost [1], it is of high temperature, high ice content, thin thickness and poor stability [2,3]. The permafrost is very sensitive to the change of ecological environment and human activities due to global warming [4–9]. Acceleration of permafrost degradation, deepening of permafrost active depth and disappearance of island permafrost are reported [10–12]. Meanwhile, construction of engineering infrastructures, such as the Qinghai–Tibetan highway, has dramatically

altered the original regime of groundwater and surface water resources. Surface runoff and roadside water has directly or indirectly affected the stability of permafrost. It bears great significance to the engineering/construction and the protection of the ecological environment that the research of the permafrost's distribution state, active depth and the fine structure is conducted.

Ground penetrating radar (GPR) is the most powerful and widely used geophysical tool in permafrost studies [13–18]. Combined with other invasive geological explorations, such as trenching, coring and boring, it is an efficient and meticulous way to study the permafrost distribution characteristics, burial location and the evolution process [10–22]. There are many reasons which affect the process of the freezing and thawing of permafrost in Tibetan Plateau [23,24], such as slope direction, slope, vegetation, the thickness of the snow cover and permafrost duration, organic layer and the soil properties, human engineering activities, etc. The thickness of permafrost layer, water content, ice content and the activity layer depth have lateral changes, obviously within the local scale, which changes the layers' electrical structure obviously. The ice, the position of the small fault and the local fine structure in the internal of permafrost, which can indicate the change of the state of internal freezing and thawing process, affect the freezing and thawing of permafrost in the context of climate change. However, it is difficult to identify the internal structure of the permafrost in complex area from the raw radar profile. In addition, the shallow permafrost layer has repeated freezing and thawing. It is difficult to effectively reflect the state of active permafrost with seasons from the radar section.

The process of migration returns the underground reflection point information back to properly positioned reflections, and the reflection wave, simultaneously with the diffraction wave, automatically converges and interferes with automatic decomposition; better migration methods can provide high resolution interpretation of imaging. Compared with Kirchhoff migration method [25,26], reverse time migration (RTM) can effectively use the full wave field information [27–29]. While compared with one-way wave equation migration [30], RTM has no limitations in propagation direction and dip angle, since RTM does not need to separate the wave field, and can better use the reversed branch and the multiple waves; it is quite adaptive to lateral velocity variations, too. RTM is the state of the art in high precision migration methods.

When the electromagnetic (EM) impulse has a central frequency well above the transition frequency (the ratio of the dielectric constant to the electric conductivity of the media) the EM filed propagation is essentially a wave field; the kinematic characteristics of the propagation of the EM wave and the elastic wave are quite similar, such that all the seismic data processing techniques can be used to process the radar data [31]. The conventional migrations have been widely used in radar data imaging [32,33]. Fisher et al. was the first to apply the finite difference (FD) RTM to GPR [31]. They simply used the acoustic wave equation to the post-stacking GPR data. Sanada and Ashida continued the work to develop the RTM algorithm directly from the Maxwell's equation with the consideration of finite conductivity in the algorithm [34]. Leuschen and Plumb realized the FD-RTM on both multi-offset and zero-offset data [35]. Zhou conducted the FD-RTM based on Maxwell's equations and achieved the common shot point prestack RTM of GPR data [36]. Liu combined the FD-RTM algorithm and the full waveform inversion to estimate migration velocity [37].

In this study, we introduced the principle of GPR and applied the FD-RTM algorithm to the GPR data acquired at the site of Beiluhe in the Tibetan Plateau. Combining the RTM imaging sections from real data and forward modeling, we demonstrated that the FD-RTM can be applied to determine the active layer depth and characterize the fine structures in the active layer in the Tibetan Plateau's permafrost regions. Through the design a series of active layer models, we generated the forward modeling synthetic data and demonstrated that RTM is a powerful tool to study the structure of the active layer of the permafrost. In the Beiluhe region the imaging result of GPR RTM profile shows that the RTM technique can clearly characterize the positions of the fine structures in the migrated section. It has great significance of using RTM section to research complex structure and freezing and thawing process in the active layer at the Beiluhe region.

## 2. Materials and Methods

### 2.1. Principles of GPR

The GPR system consists of transmitter antenna, receiver antenna, controller, data logger and display, as shown in Figure 1. Electromagnetic waves from the transmitting antenna propagate into the surrounding medium at a certain velocity which is dependent mainly on the dielectric constant of the medium [38,39]. When the electromagnetic wave encounters the interface of the medium, part of the energy is transmitted to the second layer of medium, and the remaining energy is reflected according to the reflection coefficient R which is given by:

$$R = \frac{\sqrt{\varepsilon_1} - \sqrt{\varepsilon_2}}{\sqrt{\varepsilon_1} + \sqrt{\varepsilon_2}} \tag{1}$$

where the $\varepsilon_1$ and $\varepsilon_2$ are the dielectric constants of the first layer and the second layer of the medium, respectively [40]. The reflected electromagnetic wave is received by the receiving antenna. The travel time and amplitude information of the signal can be used to image the medium.

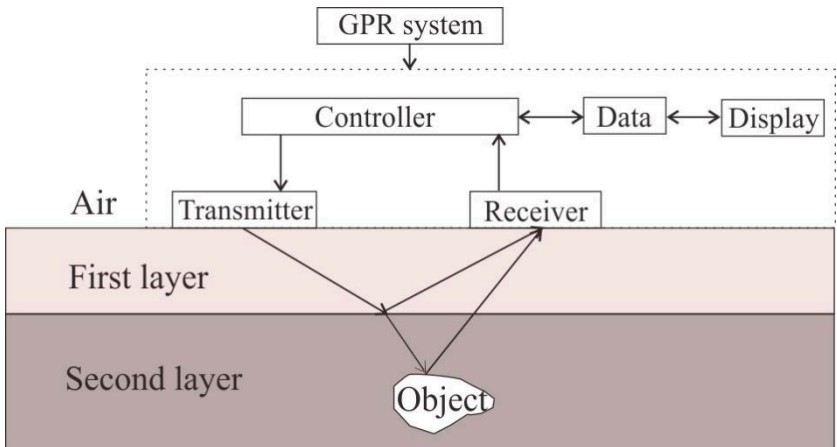

**Figure 1.** Ground penetrating radar (GPR) system and detection diagram.

### 2.2. Reverse-Time Migration (RTM)

The RTM procedure consists of three steps: forward continuation, reverse continuation and imaging. First, the forward continuation propagates the wave energy to the maximum moment along time axis; the direction of wave field and the result are preserved. Second, reverse continuation is propagated the wave field to zero moment along the time axis in reverse direction, and then, the positive wave field is read, which has same time as the forward continuation. In the third step, appropriate imaging condition is applied to get the information of underground structure. The detailed realization process is shown in Figure 2.

In the process of RTM, we choose the zero-lag cross-correlation imaging condition and the Laplace filter to remove the low frequency noise at the subsurface [41,42].

$$image(x, y) = \sum_{time} S(x, y, t) R(x, y, t) \tag{2}$$

where $image(x, y)$ is the result of imaging; $S(x, y, t)$ is the wave-filed of forward continuation in time domain, which is calculated from forward modeling data. $R(x, y, t)$ is the wave-field of reverse continuation in time domain, which is obtained from the acquired GPR data.

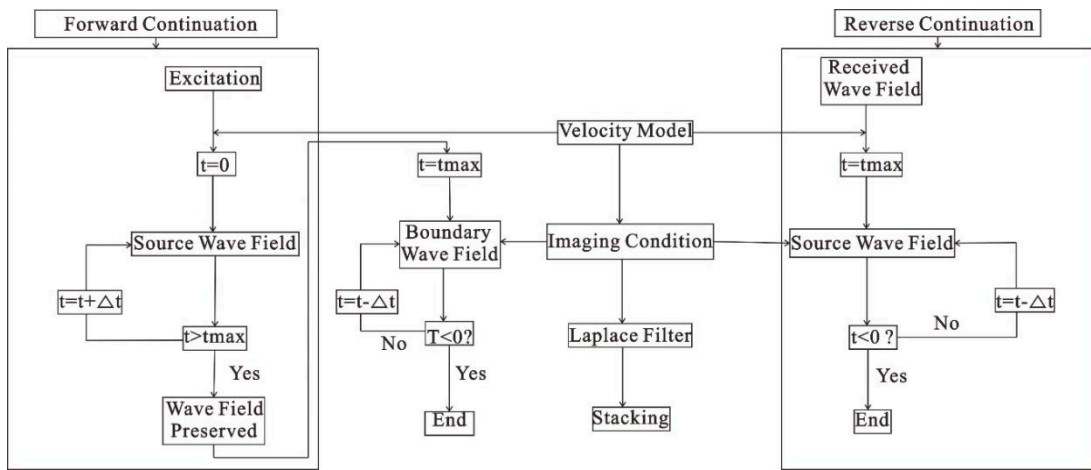

**Figure 2.** The realization process of reverse time migration (RTM) algorithm.

### 2.3. Two-Dimensional Radar Wave Forward Modeling

Efficiently solving the forward problem is the key to RTM imaging. The major task of the forward modeling algorithm is solving 2D wave equation from the original Maxwell's equations in rectangular coordinate system. The original Maxwell curl equations can be written as

$$\nabla \times H = \frac{\partial D}{\partial t} + J \tag{3}$$

$$\nabla \times E = -\frac{\partial B}{\partial t} - Jm \tag{4}$$

where *E* is the electric field; *D* is electric displacement; *H* is the magnetic field; B is the magnetic flux; *J* is electric current density; $J_m$ is magnetic electric current density. For the 2D TE (Transverse electric) mode $\partial/\partial_z = 0$ and $H_z = E_x = E_y = 0$. Therefore, in the Cartesian coordinates we have:

$$\frac{\partial E_z}{\partial y} = -\mu \frac{\partial H_x}{\partial t} - \sigma_m H_x \tag{5}$$

$$\frac{\partial E_z}{\partial x} = \mu \frac{\partial H_y}{\partial t} + \sigma_m H_y \tag{6}$$

$$\frac{\partial H_y}{\partial x} - \frac{\partial H_x}{\partial y} = \varepsilon \frac{\partial E_z}{\partial t} + \sigma E_z \tag{7}$$

where $\varepsilon$ is the dielectric constant; $\mu$ is unit permeance; $\sigma_m$ is magneto conductivity; $\sigma$ is the electric conductivity; *Hx* and *Hy* are the x and y components of the magnetic field, respectively; and *Ez* is the electric field oriented in z direction.

### 2.4. The Forward Modeling Data RTM

Since groove is a common geological structure in the field, the propagation of electromagnetic waves in groove is complicated, so a groove model is often used to verify the accuracy of radar or seismic imaging algorithms. In order to examine the correctness of the RTM algorithm, we designed a groove model (Figure 3a) for forward modeling. The dielectric constants of the upper and lower media were 2.0 and 4.0, respectively. The model size was 4 × 4 m. The grid spacing was 0.02 m. Ricker wavelet was regarded as source wavelet, for which the peak frequency was 400 MHz. The time step was $2.0 \times 10^{-11}$ s and the recording length was 50 ns. We used finite-difference forward algorithm calculate to radar profile (Figure 3b) for groove model and used RTM algorithm calculate to migration profile (Figure 3c). From the result of the migration, we can see the reflection and diffraction (the yellow

circle in Figure 3b) caused by the uneven interface in radar profile (Figure 3b). The groove interfaces (the red circle in Figure 3c) of the shape and depth of the migration profile (Figure 3c) are similar to the original velocity model (Figure 3a). The interface reflection of two intersecting axes (the yellow arrows in Figure 3c) is caused by diffraction of the jump point on the boundary at the imaging process in migration profile (Figure 3c).

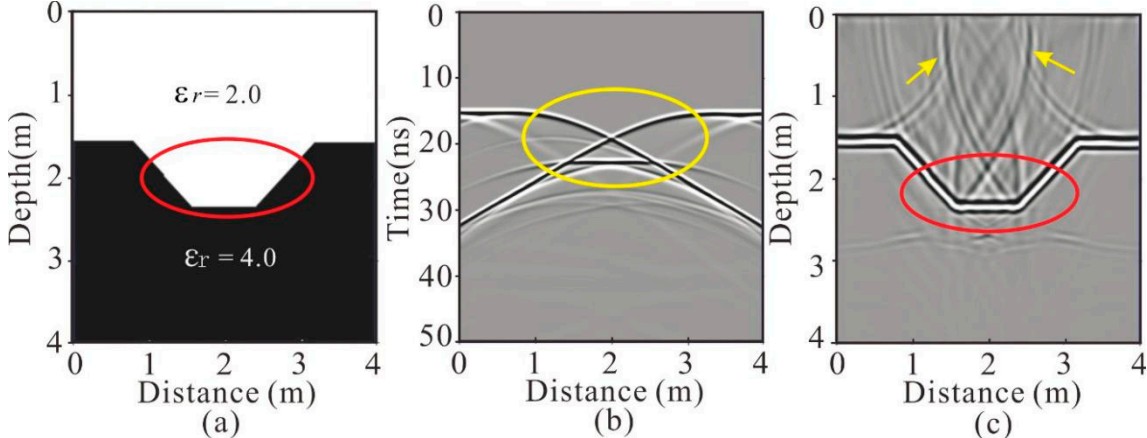

**Figure 3.** The imaging of RTM of groove model: (**a**) dielectric constant mode, (**b**) radar profile after removing the direct wave and (**c**) RTM profile.

*2.5. RTM of GPR Profiles in the Beiluhe Region*

2.5.1. Brief Description of the Study Area

The Beiluhe region (E92.9318°, N34.8214°, and see Figure 4) is located in high alluvial plain in the north part of the Tibetan Plateau. In this area, terrain is gentle and the change of elevation is from 4600 to 4700 m. The layers are mainly composed of Quaternary salt, pluvial fine sand, silt and a silty clay layer. It is covered with a peat layer in local region and made of tertiary mudstone and sandstone below 2 m. The average of annual rainfall is 300 mm, the average of annual temperature is −5.0–3.8 °C. The average of annual ground temperature is 2.0–0.5 °C in this region, respectively. Frozen soil types mainly comprise rich ice and frozen soil in the local region; the thickness of permafrost is 50–80 m; the upper limit of permafrost was −1.8 to −2.2 m over the last 30 years. The average temperatures have risen by about 0.3 to 0.4 °C in the Tibetan Plateau with the global warming. That caused degradation of permafrost region, increased the activity layer's thickness and caused the disappearance of local island permafrost [7,8,43]. The influence of human engineering activities such as the Tibetan highway, make the lithology of subsurface and coverage of vegetation is not uniform; the thickness of the active layer in the local scope has obviously changed, and the change of permafrost is transverse discontinuous.

2.5.2. GPR Data Acquisition and Processing

In May 2018, we used the GSSI SIR30 GPR apparatus for the characterization of the permafrost on both sides of the Qinghai–Tibetan highway. The GPR antennas are all bow-tie antennas, and receiver antenna and transmitter antennas are perpendicular to the survey line. We designed three survey lines with length of 200 m (the three red lines in Figure 4) and applied the common midpoint (CMP) method to obtain the formation velocity. The antennas' frequencies are 100 MHz and 400 MHz, respectively; the polarization directions of the transmitter antenna and the receiver are parallel. The sampling interval is 0.047 ns, with the total number of 1024 samples in one trace. The method of CMP is mainly used for detecting propagation velocity of radar wave. This method of profile is mainly used for detecting small structure changes at the shallow surface layer. We used 100 MHz antennas with a step size of 10 cm to move in the opposite direction of other antennas. Based on CMP velocity analysis [44,45], we measured the depth of active layer to be about 50 cm; the average found that the bottom interface

depth propagation speeds are 0.10 m/ns and 0.12 m/ns on the upper and lower parts of the bottom interface of active layer, respectively. We used 400 MHz antennas to collect high frequency GPR data along survey lines (Figure 5).

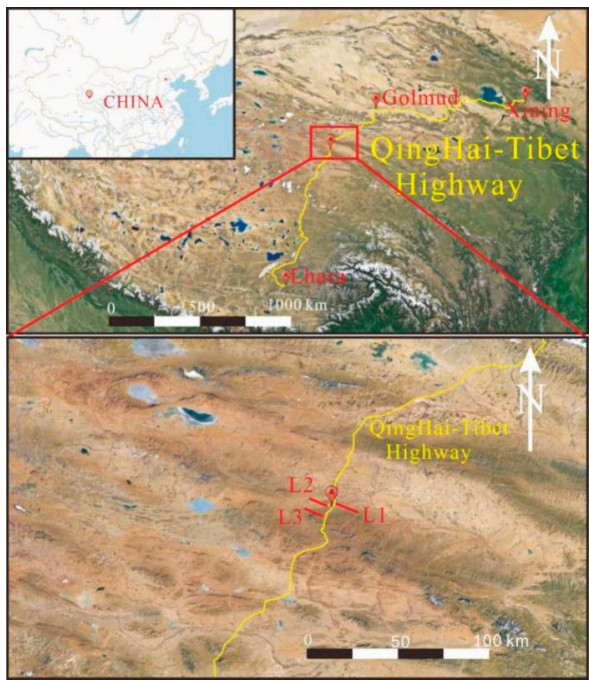

**Figure 4.** The location and lines layout diagram in Beiluhe permafrost region. Lines distribution on both sides of the Tibetan highway. We chose part of the section of on the east side of L1 line (the L1 red line) and on the west side of L3 line (the L3 red line) and L2 line (the L2 red line) to image RTM.

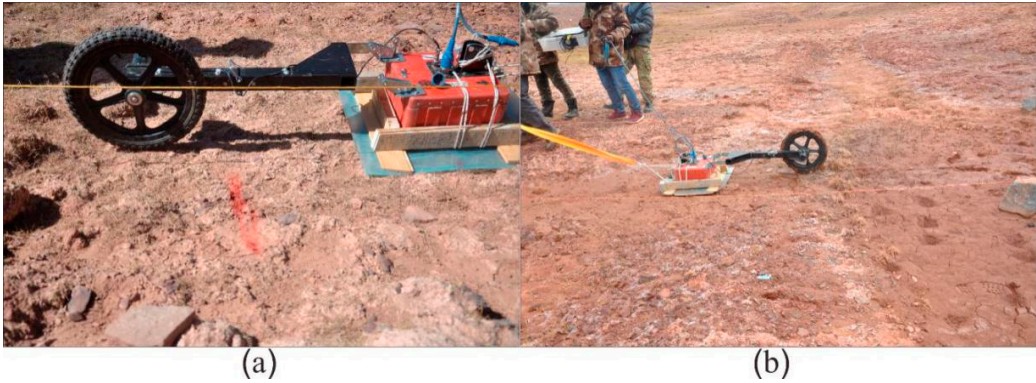

**Figure 5.** The picture of landscape: (**a**) near picture, (**b**) far picture.

When we interpreted radar data, we found they to have complex reflection and diffraction, and a small fault on the subsurface in the radar profile. In order to obtain fine structure of internal of active layer, we chose 250 trace radar datums in L1 line (the L1 line in Figure 6), 200 traces radar datums in L3 line (the L3 line in Figure 8) and L2 line (the L2 line in Figure 10) to image of RTM. In order to prove the accuracy of interpretation result of three-line RTM profiles, we designed three dielectric constant models which were based on the three-line RTM profiles and geological information of active and permafrost layers, respectively, and obtained the RTM profiles of three models.

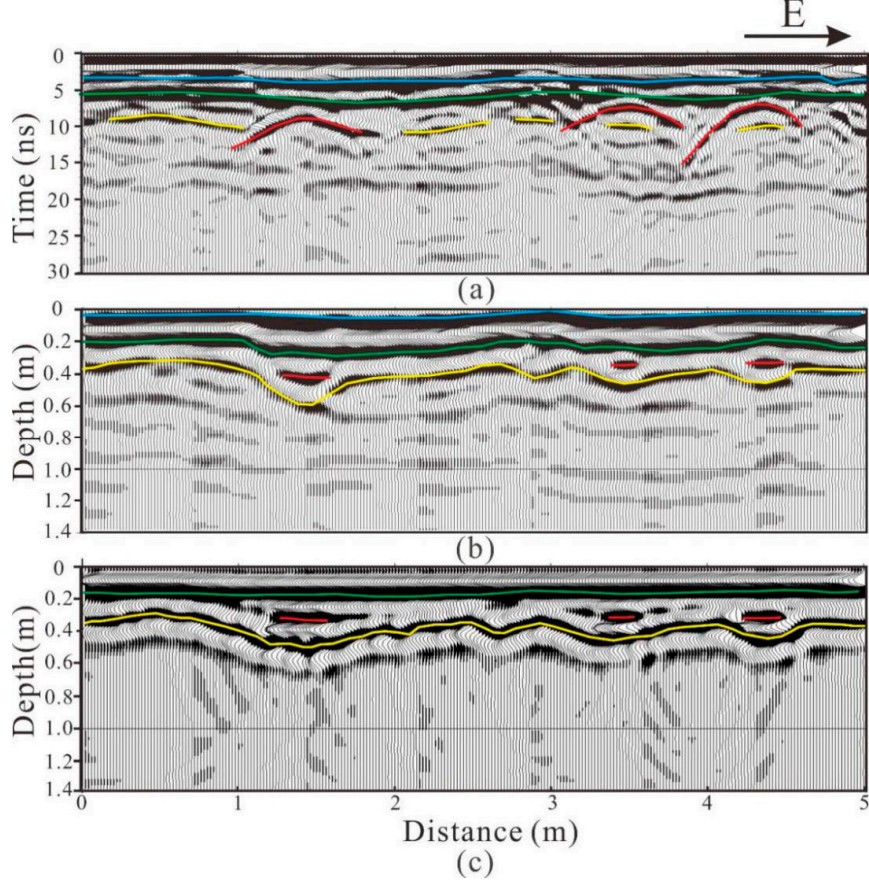

**Figure 6.** The imaging of migration of L1 line part profile. (**a**) Radar profile; (**b**) real data RTM profile; (**c**) model data RTM profile.

## 3. Results

### 3.1. Diffraction in Permafrost Internal

Figures 6 and 7 are the GPR data and dielectric constant models, respectively. The real data RTM profile in Figure 6b was calculated from the real GPR data in Figure 6a using the RTM algorithm. The dielectric constant model in Figure 7 was designed based on real data RTM profile (Figure 6b) and geological information. Then, we used the dielectric constant model to obtain forward modeling data, and calculated model data RTM profile in Figure 6c using RTM algorithm. For the sake of contrast, we put the real GPR profile, real data RTM profile and model data RTM profile into one (Figure 6). It has three obvious diffraction in the part of the permafrost profile of L1 line at the Beiluhe site (Figure 6a), which were probably caused by buried cobbles. It has a discontinuous reflection axis at 5 ns (the green line in Figure 6a), and a continuous reflection axis at 3 ns (the blue line in Figure 6a) on above of diffraction. It has intermittent reflection (the yellow line in Figure 6a), whose location at the bottom interface of the active layer is below the diffraction. Due to much attenuation, energy loss is too serious; the reflection on bottom of the energy is very weak (the yellow line in Figure 6a). It is difficult to effectively identify the position of the diffraction and determine the depth of the bottom interface of peat layer and active layer from the radar profile. The three obvious diffractions (the red lines in Figure 6a) move back to the original place after the migration (Figure 6b). The depth of the three cobbles is about 0.3 m. In combination with geological information of a shallow surface, we consider these diffractions are caused by cobbles with a size of about 10 cm. The 3 ns reflection axis (the blue line in Figure 6a) at the top of diffractions and the ground direct wave merged into a strong axis; we made sure the reflection axis was radar direct wave. The reflection axis at 5 ns on the top of diffraction

(the green line in Figure 6a) heads back to a continuous reflection axis (the green line in Figure 6b), and we consider the reflection axis as being at the bottom interface of peat layer; the depth is 0.2 m. The intermittent reflection (the yellow line in Figure 6a) heads back to continuity reflection (the yellow line in Figure 6b), and we consider the reflection the bottom interface of active layer; the depth is about 0.45 m in the radar profile. Combinations of radar migration profiles from different seasons can effectively reflect that with the climate changes, so does the process of ice changing into permafrost.

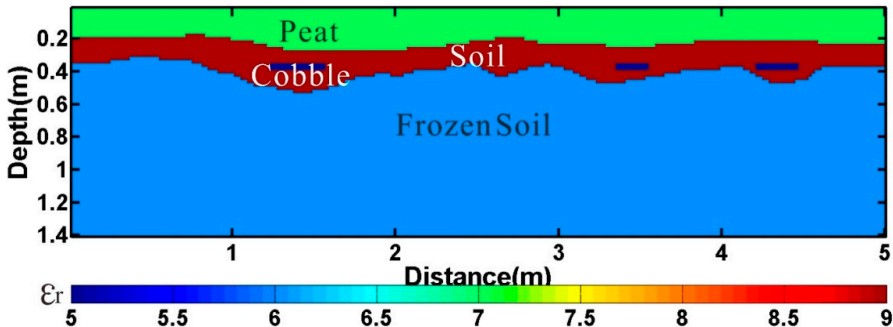

**Figure 7.** The relative dielectric constant model was designed using the L1 line RTM profile.

Based on the L1 line RTM profile, we designed a dielectric constant model of diffractions in the internal permafrost layer (Figure 7), and obtained the model data RTM profile (Figure 6c). We found exact correspondences between real data RTM profile and model data RTM profile. They have same feature and depth for the active layer (the green line in Figure 6b,c), permafrost table (the yellow line in Figure 6b,c) and the cobbles (the red line in Figure 6b,c) in the real data RTM profile and model data RTM profile.

### 3.2. Fine Structure in Internal Permafrost Layer

There are a lot of reflection axes crossing and messing in the part of the L3 line section of Beiluhe permafrost region (Figure 8a). About the time of 10 ns approximately, a very continuous reflection axis (the green lines in Figure 8a) appears, but with a lot of intermittent reflection axes (the yellow line in Figure 8a) at the top. The small reflections (the yellow line in Figure 8a) and the continuous reflection of around 10 ns (the green lines in Figure 8a) have difficulty reflecting the actual internal structure of the permafrost layer. It is difficult to reflect the fine structure of internal permafrost layer from a radar profile. The cross reflection and small reflection at subsurface (the yellow line in Figure 8a) cover the properly position, and show three distinct reflection layers (the red, yellow and green line in Figure 8b) which are significantly deeper than the depth of pit data. There are two continuous reflections on the top of active layer (the red and yellow lines in Figure 8b), which may be the interface of soil, and siltstone and fine sandstone, respectively. In the process of thawing in the RTM profile, we combine other factors with pit data and consider the depth of the bottom interface of active layer to be about 0.75 m (the green line in Figure 8b): the permafrost of subsurface, and the interface of between soil and siltstone (the red line in Figure 8b) and between siltstone and fine sandstone (the yellow line in Figure 8b) where the moisture content increased obviously, which was caused by a strong reflection in the radar profile. The depth is about 0.9 m in the RTM profile. It has weak reflection (the blue line in Figure 8b) which is caused by internal of freezing permafrost. It has an obviously lateral change the depth of interface of among the soil, siltstone, fine sandstone and active layers. Due to it having some vegetation, caused by the thawing of different levels in permafrost soil, siltstone and sandstone, it can be truly reflected internal state of freezing and thawing permafrost on the detection period.

Based on the L3 line RTM profile, we designed a dielectric constant model of fine internal structure in permafrost layer (Figure 9). Combining the geological information and characteristics of the permafrost layer, we inserted six permafrost layers in the dielectric constant model (Figure 9). After the calculation, we obtained the model RTM's profile data (Figure 8c). The layers had same characteristics

and depth of active layers and a permafrost layer (the red, yellow, green and blue line in the Figure 8b,c) in the real data RTM profile (Figure 8b).

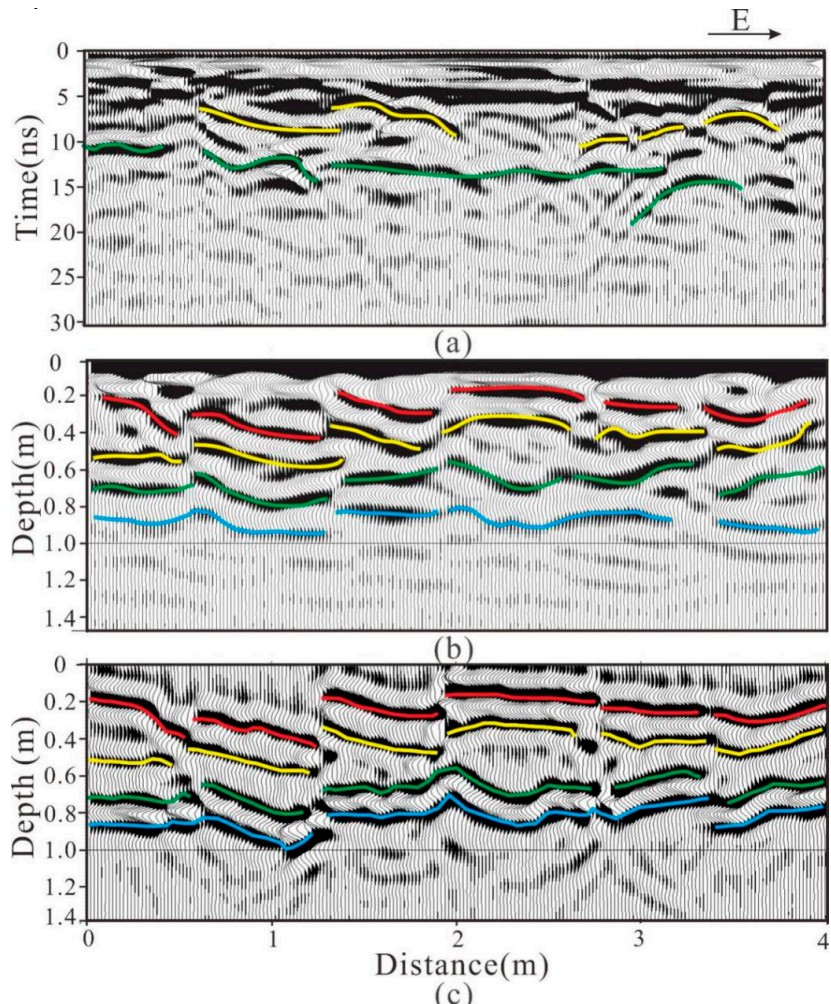

**Figure 8.** The imaging of migration of L3 line part profile. (**a**) Radar profile; (**b**) real data RTM profile; (**c**) model data RTM profile.

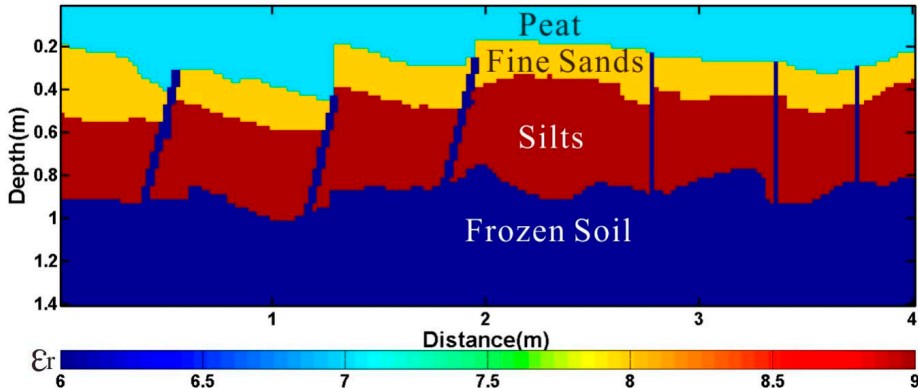

**Figure 9.** The relative dielectric constant model was designed using the L3 line RTM profile.

### 3.3. Small Lateral Fault Broken

There are two obvious small blocks at the shallow surface in part of L2 line radar profile and RTM profile (Figure 10a,b). Their depths were both about 0.2 m; both were about 5 × 10 cm. Due to

the depth being small, they may have been stone. They had a small reflection at the 5 ns in the radar profile (the green line in Figure 10a). But the energy of reflection was at the bottom of small reflection. They obviously had three reflections (the green line C in Figure 10b) which combine into one reflection in the RTM profile. That may have been interface in the frozen state. When the seasonal frozen soil layer starts thawing, the interface separates three interfaces. It has obvious convergence diffraction at the bottom of small fault (the blue line in Figure 10a), and enhances continuity reflection where the depth of reflection is 80 cm (the yellow line in Figure 10b) or 100 cm (the blue line in Figure 10b). In combination with pit data, the two reflections (the yellow and blue lines in Figure 10b) are at the bottom interface of active layer. The depth of interface is about 90 cm and deeper than the depth of the interface of the other location significantly. The layers are broken by fault F2 in the regional of E and F. Combined with geological information, the two interfaces are interfaces between soil and siltstone (the green line at regional of E and F in Figure 10b), and between siltstone and sandstone (the red line at regional of E and F in Figure 10b). They have obvious uplift at the side of up fault (D and F in Figure 10b), which indicate fault F1 and F2 are reverse faults. The formation of reverse faults is caused by unequal stress when a permafrost thaws. The region of E is risen, and on either side of the extrusion regions D and F. Due to influences of temperature, soil and shallow permafrost soil, the thawing is unequal; there are complex structures in the shallow permafrost. That can clearly reflect the position of the small fault from the RTM profile (Figure 10b). In combination with RTM profiles of different seasons, we can provide scientific advice for the Tibetan highway engineering construction.

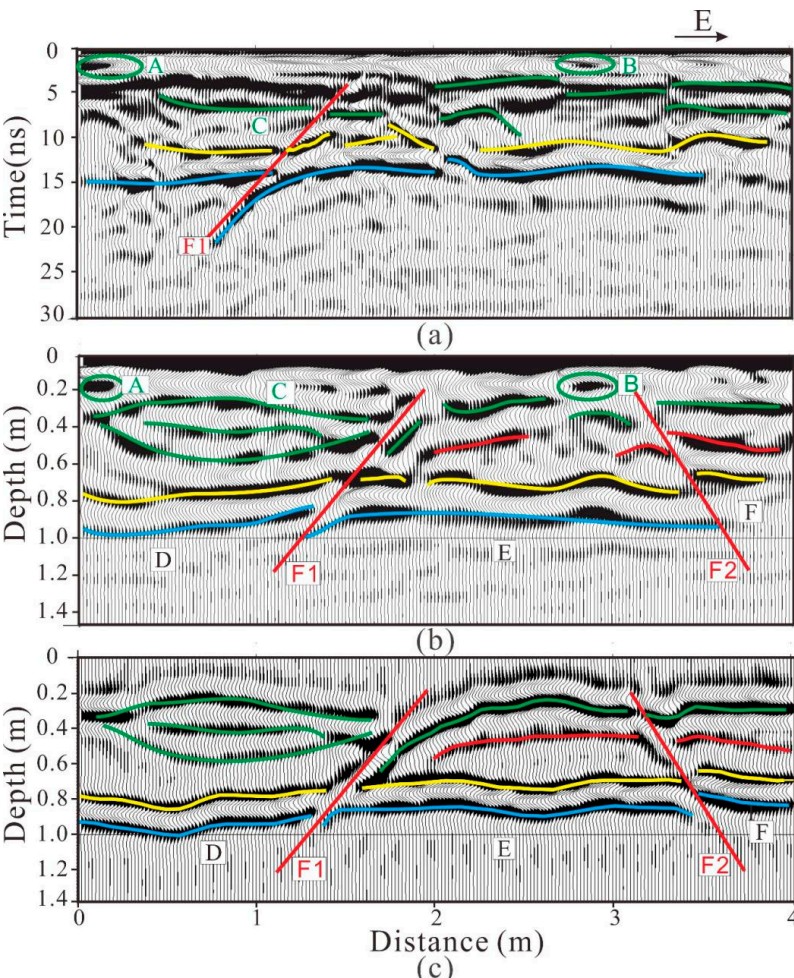

**Figure 10.** The imaging of migration of L2 line part profile. (**a**) Radar profile; (**b**) real data RTM profile; (**c**) model data RTM profile.

Based on the L2 line RTM profile and geological information of active and permafrost layers, we designed the dielectric constant model (Figure 11) of the small lateral fault, and obtained the model data RTM profile (Figure 10c). We found exact correspondence between the real data RTM profile and the model data RTM profile. They have the same features and depths for the small broken fault (F1 and F2 in Figure 10b,c) and the permafrost layer (the yellow and blue lines in Figure 10b,c).

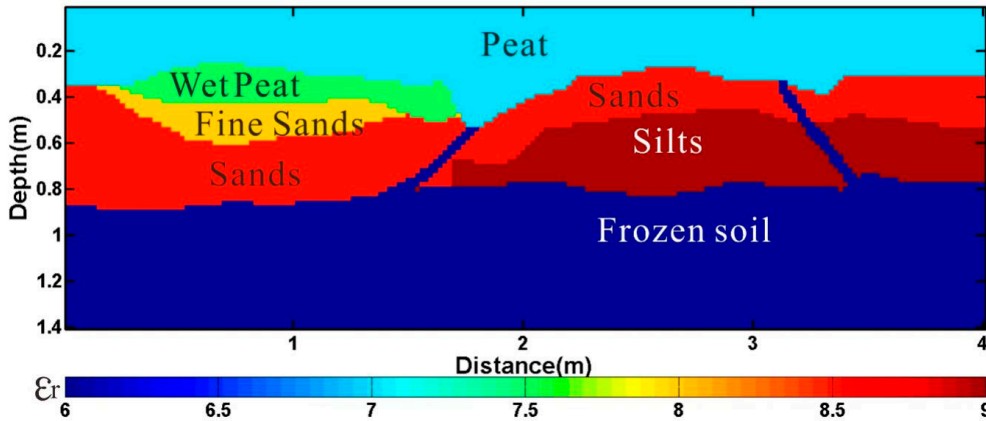

**Figure 11.** The relative dielectric constant model was designed using the L2 line RTM profile.

## 4. Discussion and Conclusions

With global warming, the Tibetan Plateau has experienced severe degradation of permafrost and disappearance of island permafrost, which has seriously affected the local ecology and stability of engineering infrastructures in the Tibetan Plateau. The place is mainly composed of a fine sand layer at the subsurface of the Beiluhe permafrost region in the Tibetan Plateau. The types of permafrost soil are mainly rich ice and frozen soil and ice containing soil at the location. Due to effects of slope, vegetation coverage and engineering activities, the active layer has obvious lateral changes in the process of freezing and thawing. There are obvious cross-overs of reflection, diffraction and energy inequality phenomena in the radar profile.

GPR provides a wealth of interpretive information about active and permafrost layer. The conductivity of the subsurface is lower (<10 mS/m) in the Beiluhe region. It has flat terrain, rarely affected by external distractions, and there is a high signal noise ratio (SNR) for radar data in this region. At the same time the CMP velocity analysis and formation depth data from pit excavation provide a precise velocity-depth model. This is the key to RTM in the Tibetan region.

Research of changes at the site of permafrost soil of Tibetan highway can provide scientific advice for the Tibetan highway's engineering construction. Previous studies have merely used the ground temperature and active layer thickness depth to research permafrost actives layer, and cannot show internal fine structure [46–48]. In this article, RTM is applied to process of permafrost GPR data of the Tibetan highway. The three RTM profiles clearly reflect the internal fine structure of permafrost and thawing state. The RTM profiles show large pieces of cobbles (Figure 6b). Due to the influence of soil at the shallow surface, the seasonal permafrost has obvious lateral changes and forms small discontinuities (Figure 8b). The depth of bottom interface of peat layer is about 0.2 m. The depth of bottom interface of the active layer has obviously changed; the deepest and shallowest parts are about 0.9 m and 0.45 m, respectively. In this region, the permafrost active layer thickness is about 1.8–2.2 m. The permafrost active layer will melt with the rise of temperature, and the maximum melting depth is in August. The survey was in May, and the maximum melting depth was about 0.9 m. Therefore, the data up to 1.4 m were sufficient.

In order to prove the accuracy of interpretation results of three-line RTM profiles (Figures 6b, 8b and 10b), we designed a dielectric constant models (Figures 7, 9 and 11) based on the three-line RTM profile and geological information, and obtained the model data RTM profiles (Figures 6c, 8c and 10c).

We found exact correspondences between real data RTM profiles and model data RTM profiles. From the three model data RTM profiles, we can prove the accuracy of interpretation results of three-line RTM profiles. In other words, the dielectric constant models of three-line (Figures 7, 9 and 11) can show the characteristics of diffraction, fine internal structure, and the small, broken lateral fault in the permafrost layer exactly.

The thawing of permafrost layer has influenced the Tibetan highway stability and indicates the change of climate. It can provide high resolution geological interpretation from the RTM profile. In combination with radar RTM profile of different seasons which can reflect the process of changing of seasonal freezing and thawing, it has great significance to researching the process of freezing and thawing of permafrost in the changing context of the Tibetan Plateau.

**Author Contributions:** Y.W., X.L., Z.F. and H.W. performed the field experiments; Y.W. and S.Q. performed the forward modeling; Y.W., X.L. and X.W. analyzed the radar data; Y.W., Z.F. and X.L. jointly wrote the paper. All authors have read and agreed to the published version of the manuscript.

**Funding:** This research was funded by the National Key R&D Program of China, grant number 2017YFC0601804; the National Natural Science Foundation of China, grant number 51777017; and the Chongqing Key Industries Common Key Technology Innovation Projects, grant number CSTC2017ZDCYZDYFX0045.

**Conflicts of Interest:** The authors declare no conflicts of interest.

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
