# Peer review of "Imaging of the Internal Structure of Permafrost in the Tibetan Plateau Using Ground Penetrating Radar"

_electronics, doi:10.3390/electronics9010056_

Round 1
Reviewer 1 Report
【Over all comment】
Experimental data using GPR and migration by RTM are being performed on permafrost on the Tibetan Plateau. Bibliographical research is mainly conducted in the field of geology, and the geological considerations obtained from the measurement results are detailed. However, the reviewers feel a little lack of commentary on the electrical and electronic parts. The process of migrating GPR data to increase resolution is generally well known. If you insist on novelty in the application of RTM, the reviewers think it is necessary to show the details and clear differences between the principle method and the conventional technology.
As described in lines 288-289, the results of providing scientific advice for the Tibetan expressway infrastructure are important field experimental data from the viewpoint of environmental issues. However, this is from a geological and ecological point of view, and reviewers do not think it to be a novelty in the field of electrical and electronics. Similarly, in lines 289-291, there is a description that the past research is simply used as the ground temperature and the thickness of the permafrost active layer and the internal detailed structure cannot be presented. Even so, it is a novelty in the field of geology, and it does not fall under the novelty in the field of electrical and electronics.
【Major comments】
Equation (1) is an equation representing an image in the xz plane. However, equations (4)-(6) represent the electromagnetic field in the xy plane. In order to avoid confusion for readers, the reviewers think it is better to unify them to either one. Although, the explanation of S(x, z, t) and R(x, z, t) in equation (1) is described in line 104, can you explain what data correspond to them in this study? In Fig. 1, scattering analysis of the groove model by FDTD is performed. But analytical condition is omitted. Can you comment the analysis conditions such as the spatial shape of the input wave source, the time pulse waveform, and the dielectric constants of the two types of media? It is described that the commercial measuring instrument GSSI SIR30 GPR is used. There are two frequencies, 100 MHz and 400 MHz. Please comment on how to use both. Please describe the polarization of the antenna, which is important for GPR. Along with this, the reviewers think it is necessary a figure explaining the measurement system or a figure showing the measurement landscape. Please comment how the model data RTM profile in Fig. 3 (c) was obtained from the real data RTM profile in Fig. 3 (b). In addition, please comment on how the model data RTM profile in Figure 3 (c) relates to the dielectric model in Figure 4. The same question applies to the relationship between Fig. 5 (c) and Fig. 6 and the relationship between Fig. 7 (c) and Fig. 8. In addition, please specify the color gradation unit at the bottom of Figure 4. The same question applies to Figures 6 and 8. All the measurement data is up to 1.4 m deep, but it seems insufficient compared to the permafrost thickness of 50-80 m and surface depth of 1.8-2.2 m described in line 143. Please comment on why data up to 1.4 m is sufficient.
【minor comments】
In lines 50-54, “Because… activities etc.” is not a single sentence surrounded by periods. Please reconsider. In the equation (3) on line 111, please add a description of J_m. In this connection, please add an explanation for σ_m in equations (4) and (5). These two values do not seem to be used for analysis of experimental data. In line 87, the active later models are considered a mistake of the active layer models. In line 278, the sentence “Due to effect of slope, vegetation coverage and engineering activities.” does not hold as a single sentence surrounded by periods. Please reconsider. In line 284, the sentence “When the frequency range is 10^5 ~ 10^9 Hz.” does not hold as a single sentence surrounded by a period. Please reconsider.Author Response
Please see the attachment.

Reviewer 2 Report
The author presented an interesting topic, “Imaging of the Internal Structure of Permafrost in Tibetan Plateau Using Ground Penetrating Radar.” The work is smoothly organized and presented; however, I have some points that need to address to improve the work: First of all, the main part of this paper is the imaging algorithm; I feel the author has not given enough attention to explain the algorithm and need to present in a better way and details. Is the presented images are for what the author design or experimental data. If it is experimental data, how can the author justify the result or compare? Is there any realistic image of the structure? Is there any simulation tool that can be used to simulated the actual scenario and compare it with the experimental results? It is a big question that needs to justify and prove. How did the author build the model in fig1, is its simulation? Not clear. In fig3 c, the author presents a “model data RTM profile.” Can you please clarify what kind of modelAuthor Response
Please see the attachment.

Round 2
Reviewer 1 Report
I confirmed that all answers to the reviewers' questions are reflected. However, the following points are suggested to be corrected in the future.
Equation (1) is newly added, but ε0 generally refers to the dielectric constant of 8.854x10^-12 F/m in air or vacuum. To avoid confusion, it is recommended to use ε1 in the first layer and ε2 in the second layer. Although the dielectric constant has been added to Figure 3 (a), the reviewer propose a clear distinction between dielectric constant ε and relative dielectric constant εr. Specifically, it is recommended to use εr = 2 and εr = 4. New measurement landscape has been added in Figure 5, but it is necessary to comment on the difference between (a) and (b). Although ε is added to the color gradation in Figure 7, it is recommended to use εr, which represents the relative permittivity. The same comment applies to Figure 9 and Figure 11.Author Response
Please see the attachment
